# The Dual Burden of Emerging Adulthood: Assessing Gambling Severity, Gambling-Related Harm, and Mental Health Challenges

**DOI:** 10.3390/ijerph21060702

**Published:** 2024-05-30

**Authors:** Belle Gavriel-Fried, Inbar Malka, Yafit Levin

**Affiliations:** 1The Bob Shapell School of Social Work, Tel Aviv University, Tel Aviv-Yafo 6997801, Israel; inbarmalka@gmail.com; 2Department of Social Work, Ariel University, Ariel 40700, Israel; yafitl@ariel.ac.il

**Keywords:** emerging adults, problem gambling severity, gambling-related harm, depression, anxiety, representative sample, Israel

## Abstract

Growing concerns over gambling problems across age groups have sparked research in public health and psychology. During emerging adulthood, individuals are more susceptible to mental health problems and more likely to develop gambling problems than in other age groups. This study explored the potential differences between emerging adults and adults aged 30+ in terms of problem gambling severity (PGS), gambling-related harm (GRH), depression and anxiety, and the mediating role of depression and anxiety in the association between age, PGS, and GRH. A representative online sample of 3244 Israelis aged 18 and over was divided into two groups: 740 emerging adults aged 18–29 and 2504 adults aged 30+. Gambling behaviors, the Problem Gambling Severity Index, the Short Gambling Harm Screen, and the Patient Health Questionnaire-4 assessing depression and anxiety were administered. Emerging adults had significantly higher levels of GRH, PGS, and depression-anxiety than their older counterparts, above and beyond gender and education. Depression-anxiety fully mediated the associations between age and gambling-related outcomes. These findings underscore the importance of considering psychological well-being in efforts to address problem gambling and gambling-related harms, especially in emerging adults.

## 1. Introduction

The increasing prevalence of gambling behaviors and gambling problems across age groups, particularly among emerging adults, is a significant public health and psychological concern [1,2,3]. Emerging adulthood is a distinct developmental stage of life between the ages 18 and 29 characterized by possibility and identity explorations, instability, self-focus, feeling ‘in between’, and changes in various life domains [4,5]. While these developmental characteristics can lead to personal growth and other positive outcomes, such as heightened self-awareness and self-knowledge, freedom, and optimism [6,7], they may also account for the high involvement of emerging adults in risk behaviors [8,9,10] and high levels of mental health problems, such as depression and anxiety, compared to other age groups [11].

Previous studies have linked gambling behaviors in young adults to later problem gambling severity [12,13] and showed that newer forms of gambling (e.g., online betting) posed greater risks [14]. This trend suggests a developmental trajectory where gambling behaviors peak during emerging adulthood, thus underscoring the need for age-specific interventions [15].

As a developmental stage of inherent instability, emerging adulthood represents a period of heightened susceptibility to mental disorders [16]. Studies have consistently shown high levels of depression and anxiety in emerging adults [17]. In a national sample of American 18- to 60-year-olds, one study showed that feelings of anxiety and depression were reported significantly more often by emerging adults [11].

Depression and anxiety are consistently associated with problem gambling [14,18]. However, the interplay between mental health and gambling is complex. Some studies have posited that this association stems from the ineffective nature of gambling to alleviate feelings of depression and anxiety [19,20]. Alternatively, individuals with problem gambling may experience heightened anxiety and depressive symptoms due to the negative consequences of their gambling behavior [20]. Hence, this association could function as both a precursor [21] and a result of problem gambling [22]. This reciprocal relationship has been confirmed in samples of emerging adults [12,23]. For example, Edgerton et al. [24] defined five different classes combining varying levels of depression and gambling. The most frequent class, accounting for 81% of the respondents (*n* = 679), consisted of individuals who experienced a gradual decline in gambling and a simultaneous increase in depression [24]. This underscores the potential of mental health symptoms to mediate the link between age and gambling behaviors.

Another account of the association between depression and anxiety symptoms and PG views these symptoms as a mediator between age-related aspects and PG. A study conducted in UK reported that in 595 individuals aged 65 and above, age-related vulnerabilities encompassing physical and psychological factors, such as clinical frailty and geriatric pain, contributed to the manifestation of PG in late-life, with depression and anxiety serving as mediators [25]. In addition, it was found that loneliness and sense of mastery as age-related factors [26,27] were associated indirectly with excessive gambling via psychological distress in four different country samples of teenagers and emerging adults [28], thus highlighting the complex effects of age and mental health on gambling.

The most widely used scale to measure problem gambling in population surveys is the Problem Gambling Severity Index (PGSI) [29]. This scale has been used in numerous epidemiological studies worldwide, e.g., [30,31,32,33], and on emerging adults, e.g., [34,35]. Recent calls have been made to measure gambling-related harms in addition to PGSI to achieve a more comprehensive picture of public health in the field of gambling [36].

Both public health theory and empirical findings indicate that individuals can experience harm from gambling without meeting the clinical criteria for gambling addiction [37]. For instance, studies have consistently shown that even gamblers classified as non-problem or at-risk gamblers on the PGSI still report harm that necessitates some form of intervention, whereas individuals classified as problem-gamblers may not report any harm [36]. Consequently, relying solely on the PGSI may not be sufficient and may overlook individuals who suffer negative consequences as a result of gambling [36]. This suggests that a dual index would be of value [38]. To the best of our knowledge, no study has explored gambling-related harm in emerging adults as a separate group.

Based on the literature review, the current study examined the differential impact of gambling behaviors in a sample of emerging adults compared to older adults by focusing on gambling harm, problem gambling severity, and depression-anxiety levels. It was hypothesized that emerging adults would exhibit higher levels of these gambling outcomes, with depression-anxiety potentially mediating the relationship between age and gambling-related consequences.

## 2. Materials and Methods

### 2.1. Sample and Procedure

This study is part of a broader research project on gambling behaviors in Israeli adults [31]. The participants were recruited through a web-based survey institution that provided a representative random sample of the Israeli population. The panel was constituted by BI and the Lucille Cohen Institute for Public Opinion Research at Tel Aviv University according to methodological standards for web-based studies defined by the National Opinion Research Center at the University of Chicago and the PEW Research Institute for studies on Israeli society [39]. The panel included a stratum for the Jewish population and a stratum for the Arab population, thus reflecting Israel’s ethnonational stratification. More details about the sample are described elsewhere [31].

Overall, the data were collected between July and September 2022, and comprised 3244 individuals: 3080 Israeli Jews and 164 Israeli Arabs. For this study, the sample was divided into two age groups consisting of individuals aged 18–29, and 30 and above. 

### 2.2. Instruments

#### 2.2.1. Gambling Behaviors

This scale was developed to encompass the 11 most prevalent forms of gambling in Israel, along with an “other” category to account for any unspecified gambling activities. Respondents rated their frequency of gambling for each type on a five-point Likert scale ranging from 1 (‘Not at all’) to 5 (‘Once a week or more’). In this study we used this scale to determine whether the individual was involved in gambling behaviors in the previous year or not. 

#### 2.2.2. The Problem Gambling Severity Index (PGSI) 

This is a nine-item scale designed to gauge problem gambling severity in general population surveys [29]. It utilizes a 4-point scoring system, where 0 signifies ‘never’ and 3 indicates ‘almost always’. Based on their scores, participants are categorized into four levels of gambling severity: 0 indicates ‘Non-problem gamblers’, scores between 1 and 2 denote ‘Low-risk gamblers’, scores ranging from 3 to 7 identify ‘Moderate-risk gamblers’, and scores of 8 or higher indicate ‘Problem gamblers’. In the current study, the PGSI demonstrated satisfactory reliability, with a Cronbach’s alpha coefficient of 0.86 and a McDonald’s omega score of 0.88. 

#### 2.2.3. The Short Gambling Harm Screen (SGHS)

This 10-item scale measures the negative consequences of excessive gambling over the previous 12 months in six domains including health, emotions, relationships, work/study, financial, and social dysfunction [37]. The SGHS was developed as a population level measure to monitor the impact of gambling on the community. Individuals respond Yes/No to each item describing a specific harm. The total score is obtained by summing the number of Yes responses. In the current study, the SGHS demonstrated satisfactory reliability, with a Cronbach’s alpha coefficient of 0.81 and a McDonald’s omega score of 0.86, indicating its effectiveness in measuring gambling-related harm within the context of this study. 

#### 2.2.4. Patient Health Questionnaire-4 (PHQ-4) 

An ultra-brief 4-item scale derived from the Patient Health Questionnaire-4 (PHQ-4) was administered in the current study to assess the severity of depression and anxiety symptoms. This scale has two components: the first two items, known as the Generalized Anxiety Disorder-2 (GAD-2), assess levels of generalized anxiety, and the latter two, referred to as the Patient Health Questionnaire-2 (PHQ-2), assess depression symptoms. Each item is scored on a four-point Likert scale ranging from 0 (‘Not at all’) to 3 (‘Nearly every day’). The cumulative scale score thus ranges from 0 to 12. This scoring approach facilitates analysis of the measure as a continuous variable by providing a more fine-grained view of anxiety and depression severity.

The PHQ-4 served as a mediator variable in this study based on the combined continuous score of all four items, representing depression-anxiety levels. A previous study utilized this 4-item measure to effectively gauge the psychological well-being of diverse populations, with an internal consistency of 0.85. It is considered to provide insights into the intensity of anxiety-depression symptoms [40]. In the current study, the scale demonstrated satisfactory reliability, with a Cronbach’s alpha coefficient of 0.81 and a McDonald’s omega score of 0.83, thus demonstrating the utility of the PHQ-4 as a unified depression-anxiety measure.

### 2.3. Analytic Strategy

We conducted a series of one-way ANOVAs and mediation analyses using the SPSS PROCESS macro. Three separate one-way ANOVAs were conducted to evaluate the impact of age categories (emerging adults vs. adults aged 30+) on the three dependent variables: the Short Gambling Harm Screen, the Problem Gambling Severity Index, and depression-anxiety levels. Two mediation analyses were conducted using the PROCESS macro in SPSS [41] to test the hypothesis that depression-anxiety would mediate the relationship between age (categorized as emerging adults and adults aged 30+) on the one hand, and gambling harm and gambling severity on the other. It is important to note that the original dataset, which was weighted by level of ethnonational affiliation (Israeli Jewish/Arab) and level of education to ensure representativeness, included individuals from diverse age groups. However, the current study included a specific subset of young individuals. Finally, we conducted additional analyses to examine whether the associations between age and gambling-related outcomes were significant, while controlling for gender and education, and to specifically test the associations between gender and education and gambling-related outcomes. We conducted two analyses of variance with covariates (ANCOVAs) to test the association between age groups on the one hand, and gambling-related harm and problem gambling severity on the other, while controlling for education and gender. 

## 3. Results

The socio-demographic characteristics of the sample in the two age groups are listed in Table 1.

Descriptive statistics were calculated for the 740 emerging adult participants and 2504 older adults. Out of the total sample, 46.7% (*n* = 346) of the emerging adults (aged 18–29) gambled in the previous year. Of these, 67.6% were classified as non-problem gamblers, 20.2% as low-risk gamblers, 9.2% as medium-risk gamblers, and 2.9% as problem gamblers. Roughly 51.1% (*n* = 1280) of the adults aged 30 and over had gambled in the previous year. Of these, 73% were classified as non-problem gamblers, 17.4% as low-risk gamblers, 8% as medium-risk gamblers, and 1.5% as problem gamblers.

### 3.1. Group Differences

The group differences between the emerging adult group and the 30+ adult group are presented in Table 2. An ANOVA indicated a statistically significant age-related difference for the Short Gambling Harm Screen scores (*F*(1, 1624) = 4.72, *p* = 0.03, partial η² = 0.003). Specifically, emerging adults (mean = 0.47, SD = 0.78) reported experiencing greater gambling harm than the older adult group (mean = 0.38, SD = 0.7).

Another ANOVA examining the Problem Gambling Severity Index showed a statistically significant difference between the two age groups (*F*(1, 1624) = 4.09, *p* = 0.043, partial η² = 0.003). Emerging adults reported higher levels of gambling severity (mean = 0.81, SD = 1.69) than the older adult group (mean = 0.63, SD = 1.4).

The third ANOVA focused on depression-anxiety levels and revealed a significant age group difference (*F*(1, 3242) = 33.99, *p* < 0.0001, partial η² = 0.01). Emerging adults had higher levels of depression and anxiety (mean = 3.84, SD = 2.69) than the older adult group (mean = 3.20, SD = 2.615). Thus, the emerging adults were more adversely affected by gambling-related harms and problems, and exhibited higher levels of depression and anxiety than the older adult group.

### 3.2. Mediation Analysis

#### 3.2.1. The Mediating Role of Depression-Anxiety in the Association between Age Group and Gambling-Related Harm

The model is presented in Figure 1. The analysis revealed a statistically significant positive association between age (emerging adulthood) and depression-anxiety (β = 0.36, *t* = 2.24, *p* = 0.025). Depression-anxiety significantly predicted gambling-related harm (β = 0.1, *t* = 0.01, *p* < 0.0001).

To evaluate the indirect effect of age on gambling harm via depression-anxiety, bootstrapping techniques were utilized. Confidence intervals were constructed based on 10,000 bootstrap samples from the original data. The results at the 95% confidence level indicated that the confidence interval for the indirect effect of age on gambling-related harm was significant (estimate = 0.04, SE = 0.02, 95% CI: 0.004, 0.0714), suggesting that depression and anxiety fully mediated this relationship such that emerging adulthood (compared to the 30+ group) was associated with greater severity of depression-anxiety, which was related to more gambling-related harm. The direct effect of age on gambling-related harm was found to be non-significant (estimate = 0.14, SE = 0.09, *t* = 1.64, *p* = 0.1, 95% CI: −0.0277, 0.3155). These findings thus highlight the pivotal role of depression-anxiety in mediating the impact of age on gambling harm.

#### 3.2.2. The Mediating Role of Depression-Anxiety in the Association between Age Group and Problem Gambling Severity

The model is presented in Figure 2. Age was significantly associated with depression-anxiety (β = 0.36, *t* = 2.24, *p* = 0.025). Depression-anxiety was found to be statistically significant in predicting problem gambling severity (β = 0.03, *t* = 4.99, *p* < 0.0001). At a 95% level of confidence, the indirect effect of age on gambling severity via depression-anxiety was significant (estimate = 0.01, SE = 0.00, 95% CI: 0.0011, 0.0251), suggesting that depression and anxiety fully mediated this relationship such that emerging adulthood (compared to the 30+ group) was associated with higher severity of depression-anxiety, which in turn was related to greater problem gambling severity. These results suggest that depression fully mediated the association between age and problem gambling severity. The direct effect was marginally significant (estimate = 0.08, SE = 0.04, *t* = 1.91, *p* = 0.06, 95% CI: −0.0022, 0.1699).

### 3.3. Additional Analyses

We examined the association between age and gambling-related harm while controlling for education and gender. There was a significant effect for age group, *F*(1, 3239) = 3.41, *p* = 0.05, above and beyond education. The main effect of gender on gambling harm was not significant, *F*(1, 3240) = 1.06, *p* = 0.303. Education was significantly associated with gambling harm, *F*(1, 3239) = 4.28, *p* = 0.039, such that lower levels of education correlated with greater gambling harm. 

We next examined the association between age and problem gambling severity while controlling for education and gender. There was a significant effect for age group, *F*(1, 3239) = 3.73, *p* = 0.05, above and beyond education. The main effect of gender on gambling severity was significant, *F*(1, 3240) = 4.93, *p* = 0.027. Men reported higher levels of problem gambling severity (M = 0.46, SD = 0.77) than women (M = 0.34, SD = 0.65). The level of education was significantly associated with problem gambling severity, *F*(1, 3239) = 9.36, *p* = 0.002, such that lower levels of education correlated with greater problem gambling severity. 

## 4. Discussion

This study investigated potential differences between age groups (emerging adulthood versus older adults), problem gambling severity and gambling-related harm, and mental health (depression-anxiety) in a representative Israeli sample. The findings revealed significant differences in problem gambling severity and gambling-related harm, as well as in depression-anxiety levels between emerging adults and older adults. Emerging adults exhibited higher levels of gambling-related harm, problem gambling severity, and depression-anxiety compared to their older adult counterparts. These findings were observed above and beyond the effects of gender and education. Furthermore, depression-anxiety fully mediated the relationship between age and gambling-related outcomes.

The group differences related to problem gambling severity, gambling-related harm, depression, and anxiety highlight emerging adulthood as a developmental stage characterized by a greater risk of experiencing gambling-related harm and problem gambling severity, as well as higher levels of depression and anxiety. These findings resonate with previous research that has linked gambling behaviors and mental health during emerging adulthood [9,22,42].

The current study contributes to this body of work by elucidating the mechanisms through which age influences problem gambling severity and gambling-related harm, and emphasizing the pivotal role of depression and anxiety. Emerging adulthood is marked by significant developmental challenges, particularly concerning mental health, which manifested in this study in the form of elevated levels of depression and anxiety that may contribute to increased gambling harm and severity [6].

The self-medication hypothesis provides a useful framework for understanding these findings. It suggests that individuals who are emotionally vulnerable may use gambling as a coping mechanism to escape negative feelings [19,43]. This aligns with other studies on young adults [44]. For this particular subgroup, gambling is a way to relieve depressive symptoms, a refuge from distressing emotions, or a magical opportunity orchestrated by the hand of fate in games of chance [43]. These findings are also in line with previous studies that have applied the self-medication hypothesis in cases of alcohol and substance use in emerging adults aged 18 to 25 [45].

Studies have also shown that a depressed mood may contribute to the formation and/or perpetuation of a gambling disorder through poor emotional regulation and maladaptive coping strategies [46]. A recent study on emerging adults indicated that difficulties in emotion regulation and coping motivations linked to gambling were associated with problem gambling behavior. The simultaneous presence of poor emotional awareness and poor emotional clarity, in conjunction with a strong inclination to alleviate or escape negative mood states (i.e., coping motivations), were shown to contribute to the risk of engaging in problem gambling [47]. Another study on a sample of students (mean age of 20.62), which also found that gambling problems were associated with depressive symptoms, suggested that difficulties in identifying emotions may serve as significant factors in predicting the likelihood and severity of gambling problems. Specifically, the inclination to overlook one’s mental states and internal emotions, given their lack of coherence or when they are perceived as unendurable or chaotic, can hamper the recognition of the adverse long-term consequences of addictive behavior [44].

Anxiety, like depression, plays a significant role in the development of gambling disorders [48], as found in the current study. Studies on the mechanisms underlying addictive behaviors in the context of alcoholism can provide valuable insights into how anxiety contributes to gambling, since alcohol briefly alleviates anxiety and seems to regulate emotions, but in fact aggravates these symptoms [49]. This leads to more recurrent and severe addictive behaviors. Consequently, individuals who experience anxiety may be more prone to severe gambling disorders [48,49]. Studies on emerging adults have reported a positive correlation between the severity of anxiety and gambling, where anxiety appeared to be associated with clinical factors within the broad range of gambling activity [48].

Another intriguing explanation that goes beyond the more classic view of gambling as a means of escaping from negative emotions has been put forward for adults. Drawing on incentive–sensitization theory proposed by Berridge and Robinson [50], Rogier et al. [46] suggested that difficulty in enjoying positive experiences may account for the association between depressive symptomatology and gambling disorder. One of the fundamental characteristics of depression is diminished responsiveness to positive emotional stimuli [51]. Thus, the high hedonic rewards related to gambling may be more satisfying in cases of depression than natural rewards [46].

This study also tested the use of a Short Gambling Harm Screen to examine gambling-related harms in emerging adults. The findings showed that the mechanisms influencing the Problem Gambling Severity Index were similar to the mechanisms detected on the Short Gambling Harm Screen. This supports the utility of both measures in capturing gambling-related harms across different age groups [52,53]. The significant differences between age groups are also consistent with a recent systematic review that unequivocally emphasized the importance of age in the context of gambling-related harm, and advocated for further investigations to better understand the distribution of harm across various age cohorts [54].

The current findings also show that while depression and anxiety can be interpreted as gambling harms [54,55], they can contribute to various other aspects of harm as well. Despite the fact that the Short Gambling Harm Screen is designed to examine the detrimental effects of excessive gambling rather than behavioral symptoms [37,38], it can be inferred that anxiety and depression contribute specifically to harm and not only to the severity of gambling. Thus, even individuals who do not meet the criteria for problem gambling [36] stand to benefit from addressing their mental health issues to mitigate gambling-related harm.

This study has several limitations. The cross-sectional design of this study limits our ability to infer causality between age, mental health, and gambling behaviors. Longitudinal studies are needed to unravel the temporal dynamics of these relationships. Further, the reliance on self-report measures introduced the potential for response bias, thus highlighting the need for a multi-method approach in future research. The unbalanced proportions between the number of emerging adults aged 18–29 (*n* = 740) and adults over age 30 (*n* = 2504) in our sample constitute another limitation. This imbalance in group sizes could have impacted the statistical power of the analyses involving age comparisons. Although a larger sample size in one demographic group may lead to more precise estimates for that group, it may not provide equally robust findings for the smaller group. Finally, our participants consisted exclusively of individuals from Israel. Given the more conservative nature of the gambling market in Israel, future studies could extend this work to more diverse markets, as well as more diverse populations and cultures. 

## 5. Conclusions and Implications

The findings point to emerging adulthood as a potentially vulnerable stage for gambling problems and gambling-related harms to emerge, and the pivotal role of depression and anxiety in driving the problematic aspects of gambling behavior. The findings have important implications for both policy and practice. They highlight the need for age-specific gambling harm reduction strategies and mental health services that address the unique challenges faced by emerging adults. The current study suggests that interventions aimed at improving mental health in this age group could be effective in reducing gambling problems and gambling-related harm. Previous research has suggested that in emerging adults, problem gambling should be seen as part of a broader syndrome necessitating a holistic approach to intervention [56] based on findings showing that the co-occurrence of both problem gambling and mental health concerns among emerging adults may be accompanied by a multitude of additional challenges, including alcohol dependence, illegal drug use [57], and suicidality [58]. These recommendations are particularly important since evidence indicates that the prevalence of gambling among emerging adults remains relatively constant during this transitional period [13].

Previous studies have pointed to the importance of implementing regular screening protocols for young adults to effectively pinpoint individuals who may be experiencing escalating gambling-related harms over time [59]. It is essential for primary care providers, as well as other healthcare, social services, and public health settings, to integrate routine assessments of gambling behaviors into their protocols to facilitate the early identification of those at risk [59,60]. The findings also shed light on the imperative need to incorporate evaluations targeting mental well-being issues as part and parcel of the screening of young individuals.

Previous research has focused on factors likely to increase susceptibility to problem gambling but have neglected the importance of protective factors in mitigating this risk [61]. It is essential for emerging adults to be able to cultivate protective factors related to problem gambling. These factors include social and emotional resources, which may be a promising strategy when addressing and potentially averting gambling-related problems, particularly in young men. Efforts should concentrate specifically on increasing awareness and involvement of family members, as highlighted in recent research [61].

The findings also suggest the value of initiating public campaigns targeting emerging adults and raising their awareness that they are a vulnerable group for gambling problems and harms, and for the risks inherent to involvement in gambling behavior itself. In addition to the importance of age, these interventions and campaigns should be sensitive and adapted to gender and differences in level of education. Young individuals, in their formative years, have the potential to benefit from preventive measures that are strategically integrated into their educational journey [3]. These measures should not only focus on addressing and reducing the risks associated with gambling problems, but should also include the development and application of comprehensive strategies designed to proactively prevent the emergence of mental health obstacles and complexities.

In terms of policy, an evaluation should be conducted by regulators and policymakers to determine the suitability of existing strategies for interventions with young adults. The results here underscore the need to specifically tackle gambling problems linked to the unique challenges faced during this crucial period of emerging adulthood. For instance, research on public policy has suggested that instituting policies in institutions of higher education could potentially mitigate various risky behaviors that are prevalent among college students [62]. These measures could include the establishment of a gambling policy, conducting campus surveys on gambling behaviors, and providing gambling-related information on counseling centers and student service websites [63]. Other policy initiatives could involve providing financial assistance to autonomous scientific research establishments to evaluate issues related to gambling, prevention, and treatment that particularly impact the emerging adult population. These could also work to enforce the industry’s minimum standards to guarantee the marketing of safer gambling products for young adults [64].

Overall, this study contributes to the growing body of literature on gambling behaviors and mental health across different age groups. It underscores the importance of considering psychological well-being in efforts to address problem gambling and gambling-related harms, especially among emerging adults.

## Figures and Tables

**Figure 1 ijerph-21-00702-f001:**
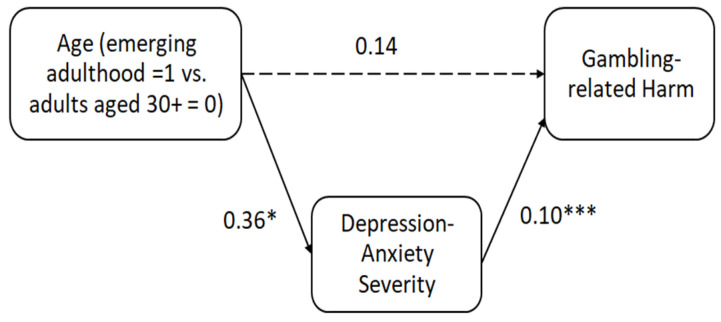
Unstandardized estimates of the indirect effects between age and gambling-related harm via depression-anxiety. Notes: *** *p*< 0.001 * *p*< 0.05. Full arrows indicate significant associations and dashed arrows indicate non-significant associations.

**Figure 2 ijerph-21-00702-f002:**
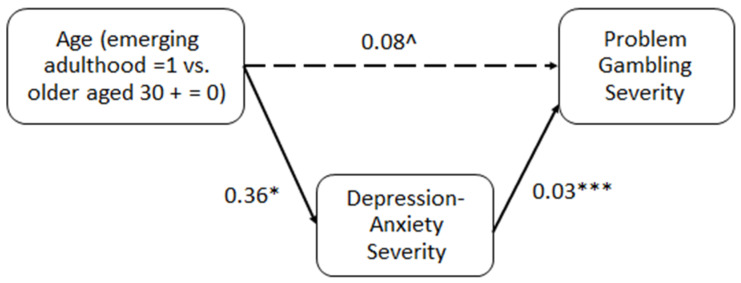
Unstandardized estimates of the indirect effects between age and gambling severity via depression-anxiety. Notes: *** *p* < 0.001, * *p* < 0.05, ^ *p* = 0.06. Full arrows indicate significant associations and dashed arrows indicate non-significant associations.

**Table 1 ijerph-21-00702-t001:** Sociodemographic characteristics according to age group.

Variable	Emerging Adults	Adults Aged 30+
Women	447 (60.4%)	1388 (55.4%)
Men	293 (39.6%)	1116 (44.6%)
In a Relationship	287 (38.8%)	1953 (78.0%)
Not in a Relationship	453 (61.2%)	551 (22.0%)
High School Diploma	316 (42.7%)	544 (21.7%)
No Formal Education	37 (5.0%)	167 (6.7%)
Partial Academic Degree	113 (15.3%)	177 (7.1%)
Full Bachelor’s Degree	179 (24.2%)	727 (29.0%)
Master’s Degree or Higher	45 (6.1%)	692 (27.6%)
Non-Academic Technical Diploma	50 (6.8%)	197 (7.9%)

**Table 2 ijerph-21-00702-t002:** Means and standard deviations for problem gambling severity, gambling-related harm, and depression and anxiety by age group.

	Emerging Adults (*n* = 740)		Adults Aged 30+ (*n* = 2504)	
	M	SD	M	SD
Problem gambling severity	0.4740	0.78068	0.3797	0.69707
Gambling-related harms	0.8092	1.69005	0.6297	1.39822
Depression and anxiety	3.8446	2.68790	3.2025	2.61578

## Data Availability

The data presented in this study are available from the corresponding author upon reasonable request.

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
