# Peer review of "The Dual Burden of Emerging Adulthood: Assessing Gambling Severity, Gambling-Related Harm, and Mental Health Challenges"

_ijerph, 2024, doi:10.3390/ijerph21060702_

Round 1

Reviewer 1 Report

Comments and Suggestions for Authors

This manuscript presents an interesting study on gambling problems among individuals of different age groups. Considering that this is a current issue, I believe it is a study of great social relevance. The introduction provides a correct contextualization of this situation, gathering and discussing studies that have focused on this problem.

In the methods section, in the instruments subsection, as a measure of reliability, the McDonald's omega could be mentioned alongside the traditional Cronbach's alpha, as it is considered a better methodological option due to its more flexible and realistic assumptions from the perspective of empirical psychological research.

Although it is important to know the differences in the variables measured by age group, it is insufficient. To analyze and understand the situation better, it would be necessary to include some additional sociodemographic variables, such as those presented in the participant description. For example, if there are differences based on participants' gender. A very interesting variable would be the level of education.

These data would help us better understand the problem and could also enrich the conclusions by helping to specify and propose prevention programs in schools, colleges, or other educational or leisure centers.

It is necessary to better explain how these results contribute to the scientific and social community, developing further what they bring to current intervention or prevention programs carried out by professionals, or what new actions could be taken.

In the limitations section, it could be mentioned that there is an unbalanced percentage between the number of emerging adult participants and the group of adults over 30 years old.

Author Response

This manuscript presents an interesting study on gambling problems among individuals of different age groups. Considering that this is a current issue, I believe it is a study of great social relevance. The introduction provides a correct contextualization of this situation, gathering and discussing studies that have focused on this problem.

Comment: In the methods section, in the instrument’s subsection, as a measure of reliability, the McDonald's omega could be mentioned alongside the traditional Cronbach's alpha, as it is considered a better methodological option due to its more flexible and realistic assumptions from the perspective of empirical psychological research.

Response: Thank you for your suggestion regarding the inclusion of McDonald's omega as a measure of reliability in the instruments subsection of the methods section. We appreciate your input and agree that McDonald's omega is indeed a valuable measure of reliability, in that it has more flexible and realistic assumptions in empirical psychological research compared to the traditional Cronbach's alpha. We incorporated the omega in our revised manuscript (please see pp. 3-4). The Cronbach’s Alpha values were updated since the analysis of the Omega considers Cronbach’s score and the factors identified.

Comment: Although it is important to know the differences in the variables measured by age group, it is insufficient. To analyze and understand the situation better, it would be necessary to include some additional sociodemographic variables, such as those presented in the participant description. For example, if there are differences based on participants' gender. A very interesting variable would be the level of education.

These data would help us better understand the problem and could also enrich the conclusions by helping to specify and propose prevention programs in schools, colleges, or other educational or leisure centers.

Response: Thank you for highlighting the importance of considering additional sociodemographic variables in our analysis. Please note that the original dataset, which was weighted based on demographics to ensure representativeness, included individuals from diverse age groups. This study analyzed a specific subset of young individuals from the larger sample.

We agree that exploring factors such as gender and education level in this young individual subset could provide valuable insights into the differences observed across age groups. We appreciate your suggestion and incorporated these variables in our analyses to enrich our findings and recommendations, please see p. 7: Additional analyses where we write:

"We examined the association between age and gambling-related harm while controlling for education and gender. There was a significant effect for age group, F(1, 3239) = 3.41, p = .05, above and beyond education. The main effect of gender on gambling harm was not significant, F(1, 3240) = 1.06, p = .303. Education was significantly associated with gambling harm, F(1, 3239) = 4.28, p = .039, such that lower levels of education correlated with greater gambling harm.

We next examined the association between age and problem gambling severity while controlling for education and gender. There was a significant effect for age group, F(1, 3239) = 3.73, p = .05 above and beyond education. The main effect of gender on gambling severity was significant, F(1, 3240) = 4.93, p = .027. Men reported higher levels of problem gambling severity (M = .46, SD = .77) than women (M = .34, SD = .65). Level of education was significantly associated with problem gambling severity, F(1, 3239) = 9.36, p = .002, such that that lower levels of education correlated with greater problem gambling severity."

Comment: It is necessary to better explain how these results contribute to the scientific and social community, developing further what they bring to current intervention or prevention programs carried out by professionals, or what new actions could be taken.

Response: We expanded the conclusion and implications sections to better explain how these results contribute to the scientific and social community. Please see p. 9-10 where we added:

"Previous studies have pointed to the importance of implementing regular screening protocols for young adults to effectively pinpoint individuals who may be experiencing escalating gambling-related harms over time [59]. It is essential for primary care providers, as well as other healthcare, social services, and public health settings, to integrate routine assessments of gambling behaviors into their protocols to facilitate the early identification of those at risk [59,60]. The findings also shed light on the imperative need to incorporate evaluations targeting mental well-being issues as part and parcel of the screening of young individuals. 

Previous research has focused on factors likely to increase susceptibility to problem gambling but have neglected the importance of protective factors in mitigating this risk [61]. It is essential for emerging adults to be able to cultivate protective factors related to problem gambling be. These factors include social and emotional resources, which may be a promising strategy when addressing and potentially averting gambling-related problems, particularly in young men. Efforts should concentrate specifically on increasing awareness and involvement of family members, as highlighted in recent research.

The findings also suggest the value of initiating public campaigns targeting emerging adults and raising their awareness that they are a vulnerable group for gambling problems and harms, and for the risks inherent to involvement in gambling behavior itself. In addition to the importance of age, these interventions and campaigns should be sensitive and adapted to gender and differences in level of education. Young individuals, in their formative years, have the potential to benefit from preventive measures that are strategically integrated into their educational journey [3]. These measures should not only focus on addressing and reducing the risks associated with gambling problems, but should also include the development and application of comprehensive strategies designed to proactively prevent the emergence of mental health obstacles and complexities.

In terms of policy, an evaluation should be conducted by regulators and policymakers to determine the suitability of existing strategies for interventions with young adults. The results here underscore the need to specifically tackle gambling problems linked to the unique challenges faced during this crucial period of emerging adulthood. For instance, research on public policy has suggested that instituting policies in institutions of higher education could potentially mitigate various risky behaviors that are prevalent among college students [62]. These measures could include the establishment of a gambling policy, conducting campus surveys on gambling behaviors, and providing gambling-related information on counseling center and student service websites [63]. Other policy initiatives could involve providing financial assistance to autonomous scientific research establishments to evaluate issues related to gambling, prevention, and treatment that particularly impact the emerging adult population. These could also work to enforce the industry's minimum standards to guarantee the marketing of safer gambling products for young adults [64]."

Comment: In the limitations section, it could be mentioned that there is an unbalanced percentage between the number of emerging adult participants and the group of adults over 30 years old.

Response: We expanded the limitations section and relate to the unbalanced percentage between the number of emerging adult participants and the group of adults over age 30. Please see p. 9: " The unbalanced proportions between the number of emerging adults (n=740) and adults over age 30 (n=2504) in our sample constitute another limitation. This imbalance in group sizes could have impacted the statistical power of the analyses involving age comparisons. Although a larger sample size in one demographic group may lead to more precise estimates for that group, it may not provide equally robust findings for the smaller group. "

Reviewer 2 Report

Comments and Suggestions for Authors

This study "examined the differential impact of gambling behaviors in a sample of emerging adults compared to older adults" from Israel.

Suggestions and questions (which may [and should] be used to improve the paper):

1. Check https://www.mdpi.com/journal/ijerph/instructions - "The abstract should be a single paragraph and should follow the style of structured abstracts, but without headings".

2. Table 1 and demographic data (lines 101-104) could be provided in the results section, since this content is part of the results.

3. Figures 1 and 2 should be cited (and explained) in the text.

4. Content related to limitations should be in the discussion section (i.e., a different section is not required). However, if the authors would like to keep this section, it could be renamed to 'Strengths and Limitations' to provide study contributions, and not only its limitations.

5. Regional generability could be discussed as a limitation, since only individuals from one country (Israel) were included in the study. Also, future work could be presented, including a study that could be conducted to determine whether the results can be generalized to other populations.

6. Avoid citations in the conclusion.

Specific points:

- abstract - line 22: problem OF gambling?

- line 55: provide reference to Edgerton et al (2018), Berridge and Robinson (2008), Rogier et al. (2019), etc. All authors should be cited.

- et al -> et al. (with a dot)

- line 116: This Thithis

Author Response

This study "examined the differential impact of gambling behaviors in a sample of emerging adults compared to older adults" from Israel.

Suggestions and questions (which may [and should] be used to improve the paper):

Comment: 1. Check https://www.mdpi.com/journal/ijerph/instructions - "The abstract should be a single paragraph and should follow the style of structured abstracts, but without headings".

Response: Thank you for your feedback on the structure of the abstract. In response to your comment, we revised the abstract to include the essential elements in a cohesive, single-paragraph format. The headings in the abstract have been omitted.

Comment 2. Table 1 and demographic data (lines 101-104) could be provided in the results section, since this content is part of the results.

Response: In the current version Table 1 is presented in the results section. See p. 4.

Comment 3. Figures 1 and 2 should be cited (and explained) in the text.

Response: Thank you for this comment. Figures 1-2 are explained in the text. See p. 4 and Page 5.

Comment 4. Content related to limitations should be in the discussion section (i.e., a different section is not required). However, if the authors would like to keep this section, it could be renamed to 'Strengths and Limitations' to provide study contributions, and not only its limitations.

Response: The limitations section has been consolidated with the discussion section, see p. 9.

Comment 5. Regional generability could be discussed as a limitation, since only individuals from one country (Israel) were included in the study. Also, future work could be presented, including a study that could be conducted to determine whether the results can be generalized to other populations.

Response: Remarks on the issue of generalizability have been incorporated into the limitations section where we write: "Finally, our participants exclusively consisted of individuals from Israel. Future studies could extend this work to more diverse populations and countries, in particular given the more conservative nature of the gambling market in Israel." (See p. 9). 

Comment 6. Avoid citations in the conclusion.

Response: Given the fact that our recommendations are based on studies that correspond with our findings, we thought that citing them would strengthen our recommendations and conclusions. Other articles published in this journal use citations in the conclusions.

Comment - Specific points:

- abstract - line 22: problem OF gambling?

- line 55: provide reference to Edgerton et al (2018), Berridge and Robinson (2008), Rogier et al. (2019), etc. All authors should be cited.

- et al -> et al. (with a dot)

- line 116: This Thithis

Response: We accepted all the comments and revised all the places in the article accordingly. Please note that “problem gambling” is a specific term, and differs from the more general notion of “the problem of gambling”. Hence, we kept the term ‘problem gambling’.